# Effect of Titanium Dioxide Nanoparticles on the Expression of Efflux Pump and Quorum-Sensing Genes in MDR *Pseudomonas aeruginosa* Isolates

**DOI:** 10.3390/antibiotics10060625

**Published:** 2021-05-24

**Authors:** Fatma Y. Ahmed, Usama Farghaly Aly, Rehab Mahmoud Abd El-Baky, Nancy G. F. M. Waly

**Affiliations:** 1Department of Microbiology and Immunology, Faculty of Pharmacy, Minia University, Minia 61519, Egypt; fatma_yousef@s-mu.edu.eg (F.Y.A.); nancy.gamil1@mu.edu.eg (N.G.F.M.W.); 2Department of Pharmaceutics, Faculty of Pharmacy, Minia University, Minia 61519, Egypt; us_farghaly@mu.edu.eg; 3Department of Microbiology and Immunology, Faculty of Pharmacy, Deraya University, Minia 11566, Egypt

**Keywords:** MDR *P. aeruginosa*, titanium dioxide nanoparticles, biofilm, efflux pumps, quorum-sensing system, real-time polymerase chain reaction

## Abstract

Most of the infections caused by multi-drug resistant (MDR) *P. aeruginosa* strains are extremely difficult to be treated with conventional antibiotics. Biofilm formation and efflux pumps are recognized as the major antibiotic resistance mechanisms in MDR *P. aeruginosa*. Biofilm formation by *P. aeruginosa* depends mainly on the cell-to-cell communication quorum-sensing (QS) systems. Titanium dioxide nanoparticles (TDN) have been used as antimicrobial agents against several microorganisms but have not been reported as an anti-QS agent. This study aims to evaluate the impact of titanium dioxide nanoparticles (TDN) on QS and efflux pump genes expression in MDR *P. aeruginosa* isolates. The antimicrobial susceptibility of 25 *P. aeruginosa* isolates were performed by Kirby–Bauer disc diffusion. Titanium dioxide nanoparticles (TDN) were prepared by the sol gel method and characterized by different techniques (DLS, HR-TEM, XRD, and FTIR). The expression of efflux pumps in the MDR isolates was detected by the determination of MICs of different antibiotics in the presence and absence of carbonyl cyanide m-chlorophenylhydrazone (CCCP). Biofilm formation and the antibiofilm activity of TDN were determined using the tissue culture plate method. The effects of TDN on the expression of QS genes and efflux pump genes were tested using real-time polymerase chain reaction (RT-PCR). The average size of the TDNs was 64.77 nm. It was found that TDN showed a significant reduction in biofilm formation (96%) and represented superior antibacterial activity against *P. aeruginosa* strains in comparison to titanium dioxide powder. In addition, the use of TDN alone or in combination with antibiotics resulted in significant downregulation of the efflux pump genes (*MexY*, *MexB*, *MexA*) and QS-regulated genes (*lasR*, *lasI*, *rhll*, *rhlR*, *pqsA*, *pqsR*) in comparison to the untreated isolate. TDN can increase the therapeutic efficacy of traditional antibiotics by affecting efflux pump expression and quorum-sensing genes controlling biofilm production.

## 1. Introduction 

According to the World Health Organization (WHO), *P. aeruginosa* is one of the most virulent and resistant bacteria for which new antimicrobials are urgently needed [1]. *P. aeruginosa* is characterized by bacterial resistance syndrome, as nearly all recognized antimicrobial resistance pathways can be found in this strain [2].

One of the important mechanisms of antibiotic resistance in *P. aeruginosa* is the exclusion of antibiotics through multidrug resistance (MDR) efflux systems, especially those that belong to the resistance-nodulation-division (RND) family. MexAB-OprM, MexXY and MexCD-OprJ are considered the main cause of intrinsic and acquired multidrug resistance [3].

Biofilm formation by *P. aeruginosa* is another mechanism of antibiotic resistance, since biofilm cells are much more resistant to antibiotics than planktonic cells [4]. The quorum-sensing system (QS) regulates the formation of biofilms and the expression of many extracellular virulence factors in many bacterial pathogens such as *P. aeruginosa* [5]. *P. aeruginosa* has four types of QS schemes, including *rhl*, *las*, *PQS*, and integrated QS (IQS). The *rhl*, *las* and *PQS* systems have been widely tested, while the IQS system was recently detected in *P. aeruginosa* [6].

Although numerous antimicrobial drugs are available commercially, they often lack efficacy against multidrug resistant (MDR) microorganisms, which is a great challenge to the health care teams [7]. Nowadays, the substitution of conventional antimicrobials with modern technology to overcome antimicrobial resistance is under trials. Nanotechnology showed promising results in many studies dealing with antibiotic resistance. Many nanostructures involving metallic particles have been created to counteract microbial pathogens [8]. Among them, titanium dioxide nanoparticles have attracted the attention of researchers due to their oxidative and hydrolytic properties [9].

Nanoparticles can exert their antibacterial activity via different mechanisms such as the following: interactions with DNA and/or the bacterial cell wall; inhibition of biofilm development; or the formation of reactive oxygen species (ROS). Nanoparticles, according to previous studies, have different modes of action than antibiotics, which may enhance their effect on MDR bacteria. [10]. This study aims to evaluate the impact of titanium dioxide nanoparticles (TDN) on the activity of the tested antimicrobials, QS and efflux pump genes expression in MDR *P. aeruginosa* isolates.

## 2. Results

### 2.1. Antimicrobial Resistance Pattern of P. aeruginosa Isolates

Twenty-five *P. aeruginosa* isolates were recovered from different clinical specimens, including three ear discharge specimens, thirteen urine, and nine wound exudate samples. Their resistance patterns were detected by the disc diffusion method against six different antimicrobial agents that are commonly used to treat *P. aeruginosa* infections.

The antimicrobial susceptibility of *P. aeruginosa* strains was detected by using the Kirby–Bauer disk diffusion method according to criteria provided by the Clinical & Laboratory Standards Institute (CLSI 2018). The *P. aeruginosa* strains were multidrug resistant (MDR) as they were resistant to three or more of the tested antibiotics (Table 1).

### 2.2. Synthesis of Titanium Dioxide Nanoparticles and Characterization

Titanium dioxide nanoparticles were prepared by the sol gel method. Dynamic light scattering was used to detect the size distribution of the formed nanoparticles on a scale that ranges from submicron to one nanometer. The average size of the formed TDN was 64.77 ± 0.14 nm (Figure 1A). The HR-TEM of TDN showed that most of the prepared TDN was spheres with asymmetrical edges and slight aggregations (Figure 1B).

X-ray diffraction (XRD) spectrum was performed to detect the phase and crystallinity of the formed TDN (Figure 1C). The XRD patterns showed that TDN was amorphous and had brookite polymorphs that were previously stated by the International Centre for Diffraction Data (ICDD card No. 015-0875).

The chemical composition of the synthesized TDN was analyzed by Fourier transform infrared spectrometry (FTIR). The FTIR spectra of TDN showed a small broad band at 1624 cm^−1^, which is characteristic to the O–Ti–O bond, a broad band at 3385 cm^−1^, which is related to O–H stretching, and two small bands observed below 1000 cm^−1^ representing Ti–O–Ti vibrations (Figure 1D).

### 2.3. Susceptibility of P. aeruginosa Isolates to TDN, TDP

The susceptibility of 25 MDR *P. aeruginosa* isolates to titanium dioxide powder (TDP) in bulk form and in nanoform was determined at different concentrations (1 μg/mL–1024 μg/mL) using the agar dilution method.

The *P. aeruginosa* strains showed high resistance to the titanium dioxide powder (TDP), as 11 isolates (44%) had MICs ≥ 1024 μg/mL and only 2 strains showed MICs ≤ 1 μg/mL. On the other hand, the titanium dioxide nanoparticles (TDN) showed excellent antimicrobial activity in comparison to TDP. As the MICs of TDN were ranged from 8 to 64 μg/mL, and three strains (12%) were sensitive to it (MIC less than 1 μg/mL) (Table 2).

### 2.4. Study of Efflux Pump System

To detect the expression of the efflux pump in MDR *P. aeruginosa* isolates, the effect of CCCP on the MICs of the tested antibiotics were determined. Table 3 showed that CCCP caused 4-fold decreases in the MICs of antibiotics in some isolates, which indicates the expression of efflux pumps in the highlighted isolates.

### 2.5. Effect of TDN on the Antimicrobial Susceptibility of the Tested P. aeruginosa

The addition of TDN resulted a in 2-fold MIC decrease or more in most of the tested isolates. As some of the antibiotics that showed no change in their MICs in the presence of CCCP showed more than a 2-fold decrease in their MICs in the presence of TDN, which suggested that TDN may have anti-efflux activity in addition to its other antimicrobial mechanisms (Table 4).

### 2.6. Characterization of Biofilm Formation Using Tissue Culture Plate Method (TCP) or Microtitre Plate Test

Out of 25 isolates, 11 (44%) isolates were high biofilm producers, 7 (28%) isolates showed moderate biofilm formation, and 7 (28%) were non or weak biofilm producers (Table 5) according to the decrease in the optical density measured in the presence of TDN in comparison to O.D in the absence of TDN.

We tested the effect of the formed TDN at a sub-inhibitory concentration (4 μg/mL) on the biofilm formation of 25 MDR *P. aeruginosa* isolates. It was found that TDN showed a high significant inhibitory effect (96%) on biofilm formation.

### 2.7. Real Time PCR

The isolates (no.2) that showed complete resistance to all tested antibiotics, presented an active efflux pump and strong biofilm producer were selected for testing the effect of TDN on the relative expression of QS genes (*lasI*, *lasR*, *rhlI*, *rhlR*, *pqsA* and *pqsR)* and efflux pump genes (*MexY*, *MexB, MexA* and *oprM*) by using real-time polymerase chain reaction (RT-PCR).

A comparison of the expression ratios of the treated and the untreated isolates was performed for each gene, as shown in Table 6. The results showed overexpression of the efflux pump genes (*MexY*, *MexB*, *MexA*) in the untreated *P. aeruginosa* isolate, which agreed with the phenotypic results, confirming the importance of an efflux pump as one of the main resistance mechanisms in MDR *P. aeruginosa*. Furthermore, the expression ratios of the QS-regulated genes (*lasI*, *lasR*, *rhlI*, *rhlR*, *pqsA* and *pqsR*) were also high in the untreated sample.

The addition of TDN resulted in significant downregulation of the efflux pump genes (*MexY*, *MexB*, *MexA*) and the QS-regulated genes (*lasR*, *lasI*, *rhll*, *rhlR*, *pqsA*, *pqsR*) in comparison to the untreated isolate. On other hand, a slight increase in the expression of *oprM* gene was shown.

## 3. Discussion

With the emergence of antibiotic-resistant bacteria like *P. aeruginosa*, the use of conventional antibiotics has contributed to the failure of treatments. Therefore, there is an urgent need for the introduction of new antimicrobial agents or the use of non-antimicrobial agents to increase the therapeutic activity of the current antibiotics [11].

The present study evaluated the antibacterial activity of TDN against MDR *P. aeruginosa* isolates. Multidrug resistant isolates represent one of the most important challenges in Egypt due to many factors including the misuse of antibiotics and biocides [12], which leads to the accumulation of antibiotic resistance and cross-resistance among antibiotics and the appearance of multidrug resistant (MDR), XRD, PDR *P. aeruginosa* [13].

The wide spread of MDR strains in Egypt in comparison to other countries warns us that strict antibiotic prescribing policies need to be implemented [14].

In the present study, twenty-five *P. aeruginosa* isolates recovered from different clinical specimens were considered as multidrug resistant isolates, as these strains were non-susceptible (resistant or intermediate) to one antimicrobial agent in three or more different antimicrobial classes [15].

The activity of TiO_2_ nanoparticles is of interest to researchers due to the unique characterization of TiO_2_ nanomaterial involving crystal size, shape, surface stability and structure [16]. Titanium dioxide nanoparticles (TDN) were prepared and characterized by DLS, HR-TEM, XRD, and FTIR. The average size of the titanium dioxide nanoparticles was 64.77 nm. The antimicrobial activities were due to the large surface area and their ability to penetrate the cell wall [17]. The prepared TDN was spheres with asymmetrical edges and slight aggregations. Nanoparticles have a strong tendency to agglomerate due to their large surface area. Typically, the agglomeration occurs due to the Van der Waals attraction forces among the nanoparticles [18]. The size and size distribution have an effect on the nanoparticle’s properties and the possible applications. Similarly, the crystal structure of the NPs and the chemical composition of nanoparticles are extensively studied [19].

X-ray diffraction peaks indicate the small size and the amorphous structure (less crystalline) of the formed TDN. As the size of nanoparticles are inversely proportional to the peak width, increases in the peak width (broad peak) indicate a small size of the formed nanoparticles and the presence of material in nonorange [20], and that supported the data of DLS. FTIR spectroscopy is used to identify the functional groups that exist on nanoparticles. The spectrum represents a fingerprint of the nanoparticles [21].

FTIR confirms the formation of TDN. The characteristic peaks for nano TiO_2_ were observed around 1624 cm^−1^, which is characteristic of the O–Ti–O bond, a broad band at 3385 cm^−1^, which is related to O–H stretching revealing the presence of water [22]; the two small bands for TiO_2_ nanoparticles observed below 1000 cm^−1^ were due to Ti–O–Ti vibrations [17]. The absence of any band at 2900 cm^−1^ indicates complete removing of absolute ethanol from the samples [23].

The MICs of titanium dioxide in bulk and nanoform were detected against *P. aeruginosa* strains. It was clear that the titanium dioxide nanoparticles had superior antibacterial activity than the bulk form, which may be due to the small size of the nanoparticles and their large surface area [24]. In addition, the antibacterial activity of TiO_2_ may be enhanced by exposure to UV light or reaction with water, leading to the generation of radical oxygen species (ROS), which have a potent oxidizing power that attacks microbial cells through various processes, such as lipid peroxidation of the cell membrane, inhibition of enzymes, damage of DNA, alteration of proteins, inhibition of enzymes, finally leading to cell death [25]. Many studies reported the antibacterial and antifungal activities of TDN [26]. However, Abdel-Fatah et al. [27] found that TiO_2_ nanoparticles had no bactericidal activity against both Gram-negative and Gram-positive bacteria isolated in a study performed in Egypt. Also, [28] they noted that titanium dioxide NPs showed very low antibacterial activity against different bacterial species, including *P. aeruginosa*.

These variations among different studies may depend on the nanoparticles’ size, concentration, particle shape, zeta potential and the tested pathogen [29]. Moreover, many environmental factors, including aeration, pH, and temperature, play a role and influence the bactericidal activity [30].

In this study, the MDR *P. aeruginosa* isolates showed a 4-fold reduction in the MICs of the tested antibiotics (ceftriaxone, ciprofloxacin, amikacin and meropenem) in the presence of CCCP. These results suggested that efflux pumps are the main mechanism of resistance in the tested isolates. Many studies from Egypt reported that the efflux pump was the main cause of the decrease in ciprofloxacin and meropenem resistance, as the MICs of ciprofloxacin and meropenem significantly reduced after the addition of CCCP [15,31].

Also, Abbas et al. [12] found that all the MDR isolates showed efflux pump activity as reported by using the ethidium bromide cartwheel method and confirmed with the presence of Mex AB-R genes by PCR.

Another study from Iran reported similar results, in which 65%, 71.5%, 60.5% and 66% of *P. aeruginosa* strains showed a significant decrease in the MICs of imipenem, cefepime, ciprofloxacin, and gentamicin, respectively, in the presence of CCCP. The inhibition of efflux pumps by CCCP in some isolates increased their sensitivity to different antibiotic classes and increased the accumulation of antibiotics that agree with our results [2]. Adabi et al. [32] noted that ciprofloxacin-resistant strains are mediated mainly by an efflux pump.

Titanium dioxide powder combined with antibiotics showed a significant decrease in the MICs of the tested strains by 2-fold or greater, which suggested its activity as an efflux inhibitor according to the method reported by Lamers et al. [33]. In addition, many studies have shown that combining TDN and antibiotics potentiates the antimicrobial action of different classes of antibiotics [34,35]. The combination of TDN with antibiotics increases the concentration of antibiotics at the bacterium–antibiotic interaction site, and the binding of bacteria to antibiotics [36]. Many studies reported that metal nanoparticles can affect proton motif force (PMF), which is essential for efflux pumps [37,38].

Chatterjee et al. [38] reported that copper nanoparticles increase the antimicrobial activity of the tested antibiotics due to the inhibition of the efflux pump of resistant *S. aureus* and *P. aeruginosa* because of copper nanoparticles on PMF.

Another significant factor leading to *P. aeruginosa* pathogenesis in clinical settings is biofilm formation [39]. Regarding biofilm results in the present study, 44%, 28%, and 28% of *P. aeruginosa* isolates were strong, moderate and weak biofilm producers, respectively.

Numerous previous studies reported various rates of biofilm formation by *P. aeruginosa* isolates. Elmaraghy et al. [40] from Egypt and Kamali et al. [41] from Iran, reported lower results than ours. While Abbas et al. [42] from Egypt reported higher results than ours.

Numerous experiments have shown the remarkable ability of metal nanoparticles to decompose thick biofilm barriers by different modes of action. The penetrating ability of metallic nanoparticles is often a valuable function for the prevention of biofilm infections [43]. In the current study, the antibiofilm activity of TDN was detected against 25 MDR *P. aeruginosa* strains. TDN showed a significant reduction in biofilm formation (96%), as TiO_2_NPs target sulfhydryl (–SH) groups in the cell membrane, resulting in the creation of the S–TiO_2_ bond, and this reaction suppresses the electron transport chain and enzymes essential for biofilm formation [24].

A recent study using TiO_2_NPs with polyvinylpyrrolidone polymer (PVP) against pathogenic bacteria on catheters showed that PVP with titanium dioxide nanoparticle films had the capability to prevent biofilm formation by *S. aureus* (83.97%) and *E. coli* (65.3%) [44]. However, Polo et al. [45] reported that neither the photocatalytic treatment with TiO_2_ film nor that with TiO_2_ nanopowder had any effect on *P. aeruginosa* biofilms at all the interfaces investigated.

Quorum sensing regulates the expression of different virulence factors and the overall process of biofilm production by pathogenic bacteria, which attracted our interest to detect the anti-QS activity of TDN at sub-inhibitory concentrations (4 μg/mL). Our results showed that TDN not only reduced the biofilm formation of *P. aeruginosa* strains, but also reduced the expressions of QS-regulated genes (*lasR*, *lasI*, *rhll*, *rhlR*, *pqsA*, *pqsR*).

Many researchers have recently began using nanotechnology for the development of nano-antimicrobials of the next generation, involving QS nano-inhibitors [46]. Silver nanoparticles were evaluated for the inhibition of QS-regulated virulence and biofilm formation in *P. aeruginosa*. AgNPs were able to decrease *LasIR-RhlIR* levels, inhibit biofilm formation, and significantly downregulate the expression of QS-regulated genes (*lasI*, *lasR*, *rhlI*, *rhlR*) [47]. Also, [48] found that glutathione-stabilized silver nanoparticles (GSH-Ag NPs) have antibiofilm activity in *P. aeruginosa* by reducing the expression the of *lasR*, *lasI* genes.

García-Lara et al. [49] found that ZnO nanoparticles decrease pyocyanin, elastase, and biofilm formation in *P. aeruginosa* strains; this indicates that ZnO nanoparticles may have QS inhibitor activity. It can be considered as an option for treatment of *P. aeruginosa* infections.

Our results also revealed that efflux pump genes (*MexY*, *MexB*, *MexA*) in an untreated *P. aeruginosa* sample were significantly overexpressed, especially the *MexY* gene. Nikaido and Pagès [50] reported that there is a noticeable relationship between MexXY-OprM overproduction in *P. aeruginosa* strains and the usage of different classes of antibiotics in treatment. It seems that MexXY-OprM has a significant benefit during antibiotic pressure in the medical setting and may play a vital role in the efflux of numerous antibiotics. Also, a direct association between the expression of the *MexA* and *MexB* genes and antibiotic resistance was reported by Dashtizadeh et al. [51], which was compatible with our findings.

In the presence of TDN, the efflux pump genes (*MexY*, *MexB*, and *MexA*) were found to be significantly downregulated. TDN may have efflux pump inhibitor activity by suppressing efflux pump genes. Abdolhosseini et al. [52] from Iran found that the efflux pump genes (*mexA* and *mexB*) in multidrug resistant *P. aeruginosa* strains were decreased by 6- and 2.75-fold, respectively, when exposed to Ag–TSC nanocomposite and ciprofloxacin. Also, Dolatabadi et al. [53] noted that both biosynthesized and commercial AgNPs downregulated the expression of the efflux pump gene *OxqAB* in resistant *Klebsiella pneumoniae* strains.

The inhibition of MDR efflux pumps by nanoparticles would be helpful in enhancing the therapeutic efficacy of traditional antibiotics by affecting quorum-sensing genes controlling biofilm production and other virulence factors [43].

## 4. Materials and Methods

### 4.1. Reagents

Titanium dioxide powder (TiO_2_, 98%) was obtained from Loba Chemie, Mumbai, India. Meropenem, ceftriaxone, cefepime powder were obtained from Pharco B international, Egypt, ciprofloxacin from Amirya, Egypt and Amikacin from Amount, Egypt.

### 4.2. Bacterial Strains

In the current study, 25 *P. aeruginosa* strains were collected from different clinical specimens (urine, ear discharge, wound exudate) of patients attending Minia university hospital. *P. aeruginosa* strains were confirmed by using the traditional microbiological method and biochemical tests [54].

### 4.3. Antimicrobial Susceptibility Testing

Antimicrobial susceptibility of *P. aeruginosa* strains was performed by Kirby–Bauer disc diffusion method according to Clinical & Laboratory Standards Institute guidelines [55]. The used antibiotics were amikacin (AK, 30 μg), cefepime (FEP, 30 μg), levofloxacin (LEV, 10 μg), ceftriaxone (CRO, 30 μg), imipenem (IPM, 10 μg) and ciprofloxacin (CIP, 5 μg).

### 4.4. Synthesis of Titanium Dioxide Nanoparticles, Characterization

Titanium dioxide nanoparticles (TDN) were synthesized as described before by Ahmed et al. [56]. Dynamic light scattering (DLS) method, high-resolution transmission electron microscopy (HR-TEM), Fourier transform infrared spectroscopy and X-ray diffraction (XRD) were used for the characterization of titanium dioxide nanoparticles (TDN).

### 4.5. Preparation of TDN Suspension

Titanium dioxide nanoparticle suspensions were prepared by adding 50 mg of TDN to 5 mL of sterile MQ water followed by shaking using ultrasound for 30 min and autoclaving at 121 °C for 20 min [39].

### 4.6. Determination of Antibacterial Activity of Titanium Dioxide Powder (TDP) and Titanium Dioxide Nanoparticles (TDN)

The antibacterial activity of titanium dioxide powder (TDP) and titanium dioxide nanoparticles (TND) were detected by agar dilution method against 25 *P. aeruginosa* strains as previously described by [31].

Briefly, overnight cultures of 25 *P. aeruginosa* strains in a Mueller Hinton Broth (MHB) were adjusted to a cell density of 10^7^ CFU/mL. Then, the bacterial culture spots were applied to the surface of Mueller Hinton Agar containing TDP and TDN with different concentrations (1 to 1024 μg/mL) using a multi-inoculator. The plates were incubated at 37 °C for 18 h and examined for growth. Spots showing no growth were defined as MIC.

### 4.7. Determination of Efflux Pumps Expression in Resistant Isolates

The MICs of ciprofloxacin, meropenem, amikacin, ceftriaxone were detected for 25 MDR *P. aeruginosa* isolates by agar dilution method in the presence and absence of efflux pump inhibitor carbonyl cyanide m-chlorophenylhydrazone (CCCP) (Sigma, San Jose, CA, USA) at a final concentration of 10 μM. A four-fold reduction in MIC or more of the tested antibiotics after adding CCCP is an indication for the presence of efflux pumps [31].

### 4.8. Effect of TDN on the Antimicrobial Susceptibility of the Tested P. aeruginosa

The MICs of ciprofloxacin, meropenem, amikacin, ceftriaxone either alone or in combination with TDN were detected for 25 MDR *P. aeruginosa* isolates by agar dilution method in the presence of titanium dioxide nanoparticles at sub-inhibitory concentrations. A two-fold or greater change in the MICs of the tested antibiotics in presence of TDN compared to MICs of antibiotics alone was reported as indicating a significant efflux pump inhibitor [33].

### 4.9. Biofilm Formation Assays

Biofilm formation of 25 *P. aeruginosa* strains was evaluated by the tissue culture plate assay method (TCP) as previously described by Christensen et al. [57]. About 1 × 10^7^ CFU ml of the tested isolates were incubated in TCP for 24 h at 37 °C. Then, bacterial cultures were removed gently and washed with phosphate buffered saline. Crystal violet (0.1%) was used to stain the adherent cells, followed by washing with PBS. Then, ethyl alcohol (95%) was applied followed by measuring absorbance at 630 nm and the results were interpreted according to Table 7. The assay was performed in triplicates for each isolate.

TDN at sub-MIC concentration was added to the tested isolates suspension in TCP and incubated for 24 h at 37 °C. Bacterial cultures were discarded and tissue culture plate wells were washed gently. The adhered cells were stained by crystal violet (0.1%) for 20 s, followed by washing using PBS. Finally, ethyl alcohol was added, and absorbance was measured as described before.

### 4.10. Gene Expression Using Real-Time PCR

The impact of TDN (4 µg/mL) on the relative expression of efflux pump genes (*MexY*, *MexB*, *MexA* and *OprM*) and quorum-sensing genes (*rhlR*, *lasI*, *lasR*, *rhlI*, *pqsA* and *pqsR*) in the chosen isolate (no.2) was evaluated using real-time polymerase chain reaction (RT-PCR).

The tested isolate was grown until the middle of the exponential phase in presence and absence of TDN. Cultures were pelleted, and mRNA was extracted using RNeasy Mini Kit (Qiagen, Hilden, Germany) according to the manufacturer’s instructions. Then, cDNA was performed using High-Capacity cDNA Reverse Transcription Kit (Thermo Fisher, New York, NY, USA).

Real-time polymerase chain reaction (RT-PCR) was performed to determine the expression level of efflux and quorum-sensing genes using Quanti Tect SYBR Green RT-PCR Kit (Qiagen, Germany) according to the manufacturer’s protocol. Synthesized cDNA, primers (Table 8) and master mix were mixed and transferred to Applied Biosystems Step One™ instrument.

The level of gene expression was relatively normalized to the expression of the housekeeping gene *rpoD*. The expression of genes in *P. aeruginosa* isolate cultivated with TDN was compared to their expression in the control cultures without TDN. Relative quantities of gene expression were calculated using the 2^−ΔΔCt^ method (Pfaffl, 2001).

## 5. Conclusions

Titanium dioxide nanoparticles can increase the therapeutic efficacy of traditional antibiotics by affecting efflux pump expression and quorum-sensing genes controlling biofilm production.

## Figures and Tables

**Figure 1 antibiotics-10-00625-f001:**
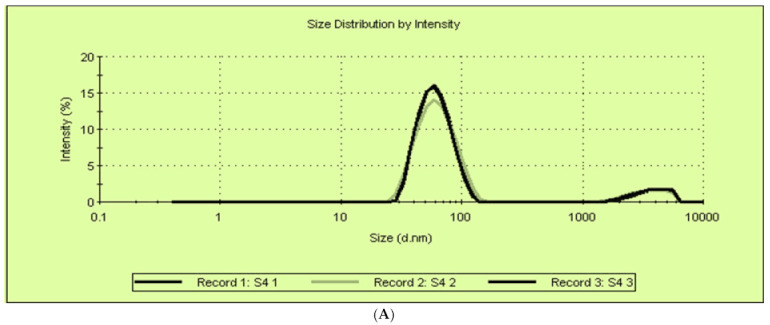
Characterization of titanium dioxide nanoparticles. (**A**) Dynamic light scattering (DLS). (**B**) High-resolution transmission electron microscopy. (**C**) X-ray differentiation. (**D**) Fourier transform infrared spectrometry.

**Table 1 antibiotics-10-00625-t001:** Antibiotic resistance patterns of the tested *P. aeruginosa* strains.

No.	Type *	CF	IM	AK	CP	LV	CT
**1**	W	R	S	R	R	R	R
**2**	W	R	R	R	R	R	R
**3**	W	R	R	R	R	R	R
**4**	U	R	R	R	R	R	R
**5**	ER	R	R	R	R	R	R
**6**	U	R	R	R	R	I	R
**7**	U	R	R	R	R	R	R
**8**	U	R	R	R	R	I	R
**9**	U	R	R	R	R	I	R
**10**	U	R	S	R	S	I	R
**11**	U	R	R	R	R	I	R
**12**	W	R	I	R	R	R	R
**13**	W	R	R	R	R	R	R
**14**	W	R	R	R	R	R	R
**15**	U	R	S	R	S	I	R
**16**	U	R	I	R	I	I	R
**17**	U	R	R	R	R	R	R
**18**	W	R	S	R	S	R	R
**19**	W	R	R	I	R	S	R
**20**	U	R	S	I	R	S	R
**21**	ER	R	R	R	R	R	R
**22**	W	R	R	R	R	R	R
**23**	U	R	R	R	S	S	R
**24**	U	R	R	R	S	S	R
**25**	ER	R	R	I	S	S	R

* Type of clinical specimen; **U**: urine specimen; **W**: wound exudate; **ER**: ear discharge. **CP**: ciprofloxacin; **CF**: cefepime; **CT**: ceftriaxone; **AK**: amikacin; **IM**: imipenem; **LV**: levofloxacin. **R**: resistance; **I**: intermediate; **S**: sensitive.

**Table 2 antibiotics-10-00625-t002:** Distribution of minimum inhibitory concentrations of TDN, TDP against MDR *P. aeruginosa* isolates (25 isolates).

Number of Isolates with MICs (μg/mL)												
	<1	1	2	4	8	16	32	64	128	256	512	≥1024
**TDN**	3	0	0	0	1	6	1	14	0	0	0	0
**TDP**	2	0	0	0	0	0	2	1	6	2	1	11

**TDN**: titanium dioxide nanoparticles; **TDP**: titanium dioxide powder (bulk form).

**Table 3 antibiotics-10-00625-t003:** Effect of addition of CCCP on antibiotic resistance pattern against 25 MDR *Pseudomonas aeruginosa* isolates.

No.	Et Br		CP		M		CT		AK	
	MIC1	MIC2	MIC1	MIC2	MIC1	MIC2	MIC1	MIC2	MIC1	MIC2
**1**	>1024	256	8	8	S	ND *	32	32	512	16
**2**	>1024	2	32	8	64	<1	1024	16	512	16
**3**	512	16	32	<1	512	<1	512	<1	128	<1
**4**	>1024	512	32	32	512	<1	>1024	16	>1024	16
**5**	>1024	512	32	2	512	<1	>1024	16	128	1
**6**	>1024	512	8	8	8	4	64	64	512	8
**7**	>1024	256	4	<1	16	8	128	128	512	16
**8**	1024	256	4	<1	16	<1	256	<1	64	<1
**9**	>1024	512	32	<1	512	<1	256	2	64	<1
**10**	512	128	S	ND *	S	ND *	256	<1	16	<1
**11**	>1024	512	8	8	16	<1	128	16	512	16
**12**	1024	512	16	<1	16	8	64	64	128	16
**13**	>1024	256	32	<1	16	8	>1024	8	1024	32
**14**	>1024	256	4	<1	16	<1	256	<1	128	<1
**15**	1024	256	S	ND *	S	ND *	256	128	16	<1
**16**	1024	512	8	<1	8	<1	32	<1	512	<1
**17**	>1024	64	8	<1	8	<1	256	4	512	<1
**18**	512	256	S	ND *	S	ND *	16	<1	512	<1
**19**	>1024	256	4	<1	64	<1	32	2	32	8
**20**	>1024	512	4	<1	S	ND *	32	<1	16	<1
**21**	1024	512	4	<1	32	2	32	8	64	<1
**22**	>1024	512	4	<1	8	<1	32	<1	512	<1
**23**	>1024	512	S	ND *	16	<1	64	16	64	<1
**24**	>1024	256	S	ND *	32	<1	64	16	64	8
**25**	256	8	S	ND *	32	<1	256	8	16	<1

**Et Br**: ethidium bromide, **CP**: ciprofloxacin; **CT**: ceftriaxone; **AK**: amikacin; **M**: meropenem, **MIC1**: antibiotic alone, **MIC2**: antibiotic + CCCP, **ND** *: not determined, **S**: sensitive.

**Table 4 antibiotics-10-00625-t004:** Effect of the addition of TDN on MICs of the tested antibiotics.

No.	CP		M		CT		AK	
	MIC1	MIC2	MIC1	MIC2	MIC1	MIC2	MIC1	MIC2
**1**	8	0.5	S	S	32	1	512	1
**2**	32	0.5	64	16	1024	512	512	64
**3**	32	1	512	256	512	1	128	1
**4**	32	8	512	256	>1024	512	>1024	1024
**5**	32	16	512	128	>1024	256	128	128
**6**	8	0.5	8	4	64	1	512	1
**7**	4	4	16	8	128	16	512	4
**8**	4	0.5	16	16	256	16	64	4
**9**	32	2	512	256	256	32	64	4
**10**	S (<1)	S (<1)	S	S	256	32	16	1
**11**	8	0.5	16	16	128	16	512	1
**12**	16	0.5	16	16	64	1	128	1
**13**	32	0.5	16	16	>1024	256	1024	4
**14**	4	0.5	16	16	256	16	128	4
**15**	S	S	s	s	256	8	16	1
**16**	8	0.5	8	8	32	1	512	1
**17**	8	0.5	8	8	256	8	512	1
**18**	**S** (<1)	**S** (<1)	**S** (<1)	**S** (<1)	16	1	512	4
**19**	**4**	**S**	64	16	32	1	32	4
**20**	4	S	S	S	32	1	16	1
**21**	4	0.5	32	16	32	16	64	1
**22**	4	0.5	8	4	32	16	512	1
**23**	S (<1)	S (<1)	16	16	64	16	64	1
**24**	S (<1)	S (<1)	32	8	64	16	64	4
**25**	S (<1)	S (<1)	32	16	256	16	16	4

**CP**: ciprofloxacin; **CT**: ceftriaxone; **AK**: amikacin; **M**: meropenem, **MIC1**: antibiotic alone **MIC2**: antibiotic + TDN, **S**: sensitive.

**Table 5 antibiotics-10-00625-t005:** Degree of biofilm formation in absence and presence of TDN among 25 MDR *P. aeruginosa* isolates.

Target	Degree of Biofilm Formation					
	High		Moderate		Non/Weak Biofilm	
	No.	%	No.	%	No.	% *
In absence of TDN	11	44	7	28	7	28
In presence of TDN	1	4	-	-	24	96

* Percentages were correlated to MDR *P. aeruginosa* isolates (25).

**Table 6 antibiotics-10-00625-t006:** Relative quantitation of gene expression of treated and control *P. aeruginosa* isolate.

Sample	Ct	ΔCt	ΔΔCt	Fold Difference of Gene Expression
**Target rpoD**		-	-	-
**(Housekeeping Gene)**				
**Ps ***	37.203			
**Ps1 ****	34.0287			
**Target lasI**			−4.1318	17.607
**Ps ***	27.9	−9.253		
**Ps1 ****	20.637	-13.3911		
**Target lasR**			−3.5783	11.9447
**Ps ***	27.721	−9.4821		
**Ps1 ****	20.9683	−13.0604		
**Target MexA**			−5.599	48.47
**Ps ***	30.7584	−6.4446		
**Ps1 ****	21.984	−12.0438		
**Target MexY**			−7.458	175.86
**Ps ***	34.5908	−2.6122		
**Ps1 ****	25.9582	−10.0705		
**Target Mex B**			−6.0032	64.1428
**Ps ***	29.905	−7.2525		
**Ps1 ****	20.773	−13.2557		
**Target OprM**			3.937	0.0653
**Ps ***	34.9282	−2.2749		
**Ps1 ****	35.6917	1.663		
**Target pqsA**			−1.499	2.8279
**Ps ***	26.669	−10.5335		
**Ps1 ****	21.99	−12.033		
**Target pqsR**			−3.9985	15.98
**Ps ***	27.9577	−9.2453		
**Ps_1_ ****	20.7849	−13.243		
**Target rhlR**			−3.2054	9.2242
**Ps ***	27.3754	−9.8277		
**Ps1 ****	20.9956	−13.0331		
**Target rhll**			−2.0059	4.0165
**Ps ***	27.1394	−10.0637		
**Ps1 ****	21.959	−12.0696		

**PS ***: Treated isolate with TDN, **PS1 ****: control isolate, **Ct**: cycle threshold, **ΔCt**: target gene Ct of sample—housekeeping gene C_t_ of the same sample. **ΔΔCt:** ΔCt of treated strain−ΔCt of control.

**Table 7 antibiotics-10-00625-t007:** Classification of bacterial adherence and biofilm formation by TCP method.

Biofilm Formation	Adherence	Mean OD Values
Non/Weak	Non/Weak	<0.120
Moderate	Moderate	0.120–0.240
High	Strong	>0.240

**Table 8 antibiotics-10-00625-t008:** The sequence of the primers used in this study.

Gene	Primer Direction	Primer Sequence	Size of Amplified Product (bp)	Reference
*rpoD*	F R	5-GCGAGAGCCTCAAGGATAC-3 5-CGAACTGCTTGCCGACTT-3	131	(El-Shaer et al., 2016)
*MexY*	F R	5-CCGCTACAACGGCTATCCCT-3 5-AGCGGGATCGACCAGCTTTC-3	246	(Yoneda et al., 2005)
*MexA*	F R	5′ACCTACGAGCCGACTACCAGA-3′ 5′GTTGGTCACCAGGGCGCCTTC-3′	179	(Pourakbari et al., 2016)
*MexB*	F R	5-GTGTTCGGCTCGCAGTACTC-3 5-AACCGTCGGGATTGACCTTG-3	244	(Pourakbari et al., 2016)
*OprM*	F R	5-CCATGAGCCGCCAACTGTC-3 5-CCTGGAACGCCGTCTGGAT-3	205	(Pourakbari et al., 2016)
*Las I*	F R	5-CGCACATCTGGGAACTCA-3 5-CGGCACGACGATCATCATCT-3	176	(El-Shaer et al., 2016)
*Las R*	F R	5-CTGTGGATGCTCAAGGACTAC-3 5-AACTGGTCTTGCCGATGG-3	133	(El-Shaer et al., 2016)
*rhII*	F R	5-GTAGCGGGTTTGCGGATG-3 5-CGGCATCAGGTCTTCATCG-3	101	(El-Shaer et al., 2016)
*rhIR*	F R	5-GCCAGCGTCTTGTTCGG-3 5-CGGTCTGCCTGAGCCATC-3	160	(El-Shaer et al., 2016)
*Pqs A*	F R	5-GACCGGCTGTATTCGATTC-3 5-GCTGAACCAGGGAAAGAAC-3	74	(El-Shaer et al., 2016)
*pqsR*	F R	5-CTGATCTGCCGGTAATTGG-3 5-ATCGACGAGGAACTGAAGA-3	142	(El-Shaer et al., 2016)

## Data Availability

Data are applicable from authors.

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
