# Peer review of "Effect of Titanium Dioxide Nanoparticles on the Expression of Efflux Pump and Quorum-Sensing Genes in MDR Pseudomonas aeruginosa Isolates"

_antibiotics, 2021, doi:10.3390/antibiotics10060625_

Round 1

Reviewer 1 Report

In this study Authors evaluate the impact of Titanium Dioxide Nanoparticles (TDN) on the activity of some antibiotics, QS and efflux pump genes expression in different isolates of multi drug resistant P. aeruginosa .

Line 22: from the first lines of the abstract and then throughout the text the acronym "CCCP" is used without specifying its meaning, please insert the meaning in the first use of line 22.

line 70: to improve the readability of the phrase it would be better to write the numbers all in the same style, that is, always using letters or alternatively numbers.

Table-1: here it would be appropriate to mention which “reference method” was used to determine the MIC values and the consequent categorizations of susceptible/intermediate/resistant.

Line 122: Addition of TDN resulted in 2-fold MICs decrease or more.

Figure-2: the quality of the image is not clear (and the same is also for Figure-1); furthermore, figure-2 is not clear even for the way in which the values "ps" are represented with respect to those of "ps1", to make the tower graph more legible, try to change the scale of the represented values.

Line 233: in The MICs of the tested by 2 folds or greater… rewrite: in the MICs of the tested strains by 2 folds or greater

Line 243: P._aeruginosa

Line 345: P. aeruginosa: use italics.

Line 358: at 24h: replace with: for 24h

Line 358: Ethyl alcohol was applied… here, specify ethanol concentration and contact time.

Line 358: During washing with PBS, usually the Pseudomonas biofilm tends to detach easily from the surfaces, it is necessary to carry out at least 5 independent tests to verify if the decrease in the biofilm volume is not due to an accidental leaching or actually to the action of the TDN; or use a more reliable method.

Table-6: here the Authors report results both on the thickness of the biofilm and on the levels of adhesion, this difference is not dealt with anywhere in the text, modify the table or add missing descriptions in the text.

Author Response

Answer

Reviewer 1

Comment

Done

Line 22: from the first lines of the abstract and then throughout the text the acronym CCCP; is used without specifying its meaning, please insert the meaning in the first use of line 22.

Done

Line 70: to improve the readability of the phrase it would be better to write the numbers all in the same style, that is, always using letters or alternatively numbers.

Antimicrobial susceptability of the tested isolates was detected by using the Kirby- Bauer disk diffusion method according to criteria provided by the clinical Laboratory Standard Institute (CLSI 2018)

Table-1: here it would be appropriate to mention which “reference method” was used to determine the MIC values and the consequent categorizations of susceptible/intermediate/resistant.

Done

Line 122: Addition of TDN resulted in 2-fold MICs decrease or more

This Figure was performed by StepOne software analysis version 3.1

So we can add the table containing  Ct  and ΔCt  as a supplementary data for the figure

(attached below)

Figure-2: the quality of the image is not clear (and the same is also for Figure-1); furthermore, figure-2 is

 not clear even for the way in which the values "ps" are represented with respect to those of "ps1", to

make the tower graph more legible, try to change the scale of the represented values.

Done

Line 233: in The MICs of the tested by 2 folds or greater… rewrite: in the MICs of the tested strains by 2 folds or greater.

Done

Line 243: P._aeruginosa.

Done

Line 345: P. aeruginosa: use italics.

Done

Line 358: at 24h: replace with: for 24h.

Done

Line 358: Ethyl alcohol was applied… here, specify ethanol concentration and contact time.

PBS osmolarity and pH makes it a physiological buffer that will not harm the cells. In addition, The assay was performed in triplicates for each strain.

Line 358: During washing with PBS, usually the Pseudomonas biofilm tends to detach easily from the surfaces, it is necessary to carry out at least 5 independent tests to verify if the decrease in the biofilm

volume is not due to an accidental leaching or actually to the action of the TDN; or use a more reliable method.

·         The results were evaluated according to the decrease in the optical density measured in presence of TDN in comparison to O.D in absence of TDN (According to Christensen et al., 1985)

·         Christensen, Gordon D, W Aꎬ Simpson, JJ Younger, LM Baddour, FF Barrett, DM Melton, and EH Beachey. "Adherence of Coagulase-Negative Staphylococci to Plastic Tissue Culture Plates: A Quantitative Model for the Adherence of Staphylococci to Medical Devices." Journal of clinical microbiology 22, no. 6 (1985): 996-1006.

Table-6: here the Authors report results both on the thickness of the biofilm and on the levels of adhesion this difference is not dealt with anywhere in the text, modify the table or add missing descriptions in the text

Reviewer 2 Report

In this manuscript Ahmed et al. characterized the inhibitory effects of titanium dioxide nanoparticles (TDP) against multi-drug resistant (MDR) isolates of P. aeruginosa. An overall observation is that TDP exhibit strong inhibitory effects against MDR P. aeruginosa  strains that seem to strong express efflux genes. One drawback here is that expression of efflux genes in the MDR isolates is indirectly inferred through the CCCP assay instead of looking at expression of genes directly. Another weakness lies in the poor description of results and the low-resolution figures which make it impossible to read. Moreover, some experiments should include proper controls. The author should correct the errors described below and perform a thorough revision of the manuscript to:

  1. Line 22: Please correct font for ‘formation’
  2. Lines 62-64: ‘According to…MDR bacteria.’ Please correct sentence structure.
  3. Line 72: Please correct ‘that commonly’ to ‘that are commonly.’
  4. Lines 73-74: Please explain in this section or in methods, what measurements of zones of inhibitions are used for classifying resistant, intermediate and sensitive phenotypes.
  5. Figure 1: It is impossible to read anything in this figure and needs to be completely redone to make it legible.
  6. Figure 1D: Needs a control for comparison.
  7. Table 2: TDB should be TDP.
  8. Section 2.4: The data in this section is missing the crucial control of showing the effect of CCCP alone. What is the MIC of CCCP against these strains?
  9. Since, these strains are multi-drug resistant it is expected that all these strains will express efflux genes. Adding an ionophore such as CCCP will obviously increase sensitivity of these strains to different antibiotics. The overall reasoning for this CCCP assay is unclear.
  10. Table 3: It is unclear what the authors mean by sensitive in the context of MIC? Either the MIC is below or above the range of the tested concentrations.
  11. Section 2.5 and Table 4: This is missing the important control of showing the MIC for TDN against each of these strains. Based the information from table 2, sub-inhibitory concentrations must have been different for each of these strains? Was the sub-inhibitory concentration of TDN for each of these strain the same?
  12. Line 202-206: Please correct sentence and spelling errors.
  13. Line 207-211: Please reference author names properly here and throughout.

Author Response

Answer

Comment

Done

 Lines 62-64: ‘According to…MDR bacteria.’ Please correct sentence structure.

Done

Lines 72: Please correct ‘that commonly’ to ‘that are commonly

Antimicrobial susceptibility of P. aeruginosa strains was detected by using the Kirby- Bauer disk diffusion method according to criteria provided by the clinical Laboratory Standard Institute (CLSI 2018)

Lines 73-74: Please explain in this section or in methods, what measurements of zones of inhibitions are used for classifying resistant, intermediate and sensitive phenotypes

The figure was modified

Figure 1: It is impossible to read anything in this figure and needs to be completely redone to make it legible

The results were compared to the following references. These references did not use a control, but they undergo FT-IR depending on that the functional characteristics of TiO2 were determined by FT-IR spectroscopy in the range of 400–4,000 cm−1 using a Shimadzu model 8300 spectrophotometer by using a KBr pellet technique (0.1 weight%).

 References

 Chellappa M, Anjaneyulu U, Manivasagam G, Vijayalakshmi U. Preparation and evaluation of the cytotoxic nature of TiO2 nanoparticles by direct contact method. Int J Nanomedicine. 2015;10(Supplement 1 Challenges in biomaterials research):31-41
https://doi.org/10.2147/IJN.S79978

Hamadanian M, Reisi-Vanani A, Majedi A. Sol-gel preparation and characterization of Co/TiO2 nanoparticles: application to the degradation of methyl orange. J Iran Chem Soc. 2010;7(suppl 2):S52–S58..

Mohan L, Durgalakshmi D, Geetha M, Sankaranarayan TSN, Asokamani R. Electrophoretic deposition of nanocomposite (Hap+TiO2) on titanium alloy for biomedical applications. Ceram Int. 2012;38(4):3435–3443

Figure 1D: Needs a control for comparison

Done

Table 2: TDB should be TDP

The antibacterial activity of CCCP ( at final concentration of 10 μM) were detected against the tested strains . CCCP had no effect on growth of tested strains.

Section 2.4: The data in this section is missing the crucial control of showing the effect of CCCP alone. What is the MIC of CCCP against these strains?

CCCP assay is used to detect presence of  efflux pumps in resistant strains (phenotypic detection) 

Since, these strains are multi-drug resistant it is expected that all these strains will express efflux genes. Adding an ionophore such as CCCP will obviously increase sensitivity of these strains to different antibiotics. The overall reasoning for this CCCP assay is unclear

Sensitive ≡ MIC is below than 1μg/ml

Table 3: It is unclear what the authors mean by sensitive in the context of MIC? Either the MIC is below or above the range of the tested concentrations

TDN was used at 4μg/ml which is considered as  sub-inhibitory concentration for all  tested strains.

Section 2.5 and Table 4: This is missing the important control of showing the MIC for TDN against each of these strains. Based the information from table 2, sub-inhibitory concentrations must have been different for each of these strains? Was the sub-inhibitory concentration of TDN for each of these strain the same?

Done

Line 202-206: Please correct sentence and spelling errors.

References were revised

Line 207-211: Please reference author names properly here and throughout.

Round 2

Reviewer 1 Report

If you intend to continue studying biofilms, just a suggestion for the future: the old method of measuring biofilm through the use of crystal violet (CV), although accepted by all journals, is now an old method of little scientific value, not repeatable with precision.  In fact: it only measures the volume of the matrix and does not distinguish the number of live bacteria from dead ones, it is particularly ineffective if you work with biofilms such as Pseudomonas, which notoriously produces a more mucilaginous and inconsistent biofilm than other microorganisms. New, more reliable, and accurate methods are emerging. Scientific journals and reviewers should begin to look suspiciously at all scientific experiments that use only CV as a biofilm study method.

Author Response

If you intend to continue studying biofilms, just a suggestion for the future: the old method of measuring biofilm through the use of crystal violet (CV), although accepted by all journals, is now an old method of little scientific value, not repeatable with precision.  In fact: it only measures the volume of the matrix and does not distinguish the number of live bacteria from dead ones, it is particularly ineffective if you work with biofilms such as Pseudomonas, which notoriously produces a more mucilaginous and inconsistent biofilm than other microorganisms. New, more reliable, and accurate methods are emerging. Scientific journals and reviewers should begin to look suspiciously at all scientific experiments that use only CV as a biofilm study method.

Thanks for your advice, I will replace this method with a newer in the future

Reviewer 2 Report

The changes made by the author have improved the flow of the main text of the manuscript. However, the presentation of the figures are still quite poor. Specially, labels of figure 2 are still very hard to read and gene names are not properly written in lower case and italics. Moreover, the manuscript still requires a final proofread to fix grammatical and spelling errors.

Author Response

Answer

Reviewer 2

Comment

We replaced Figure 2 with Table 6.

The genes’ name was revised

The changes made by the author have improved the flow of the main text of the manuscript. However, the presentation of the figures are still quite poor. Specially, labels of figure 2 are still very hard to read and gene names are not properly written in lower case and italics. Moreover, the manuscript still requires a final proofread to fix grammatical and spelling errors.